# The Cumulative Formation of R-loop Interacts with Histone Modifications to Shape Cell Reprogramming

**DOI:** 10.3390/ijms23031567

**Published:** 2022-01-29

**Authors:** Hanshuang Li, Chunshen Long, Yan Hong, Lemuge Chao, Yong Peng, Yongchun Zuo

**Affiliations:** 1State Key Laboratory of Reproductive Regulation and Breeding of Grassland Livestock, College of Life Sciences, Inner Mongolia University, Hohhot 010070, China; lhshuang@mail.imu.edu.cn (H.L.); cslong@mail.imu.edu.cn (C.L.); 31709150@mail.imu.edu.cn (Y.H.); 32008182@mail.imu.edu.cn (L.C.); 2Department of Biochemistry and Molecular Biology, The University of Chicago, 929 East 57th Street, Chicago, IL 60637, USA; yongpeng@uchicago.edu

**Keywords:** R-loops, histone modifications, transcription regulation network, somatic cell reprogramming, cell fate transitions

## Abstract

R-loop, a three-stranded RNA/DNA structure, plays important roles in modulating genome stability and gene expression, but the molecular mechanism of R-loops in cell reprogramming remains elusive. Here, we comprehensively profiled the genome-wide landscape of R-loops during cell reprogramming. The results showed that the R-loop formation on most different types of repetitive elements is stage-specific in cell reprogramming. We unveiled that the cumulative deposition of an R-loop subset is positively correlated with gene expression during reprogramming. More importantly, the dynamic turnover of this R-loop subset is accompanied by the activation of the pluripotent transcriptional regulatory network (TRN). Moreover, the large accumulation of the active histone marker H3K4me3 and the reduction in H3K27me3 were also observed in these R-loop regions. Finally, we characterized the dynamic network of R-loops that facilitates cell fate transitions in reprogramming. Together, our study provides a new clue for deciphering the interplay mechanism between R-loops and HMs to control cell reprogramming.

## 1. Introduction

R-loop is a special three-stranded nucleic acid structure composed of a DNA:RNA hybrid double helix and a single-stranded DNA (ssDNA) molecule, which is ubiquitous in the whole organism. R-loops normally form during transcription [1], and were first found in the F1 phage in 1967 [2], and were subsequently characterized in *Escherichia coli* [3]. In mammals, the predominant R-loop formation shows a strong sequence preference, which usually occurs at multiple loci with high guanine–cytosine (GC) content, including the unmethylated CpG island [4] and strong G/C skew [calculated as (G − C)/(G + C)] promoters [5]. The formation events of R-loops also occur in conserved regions that are associated with specific epigenetic modification marks [6,7]. Several studies have revealed that R-loops play a part in many biological processes, such as DNA replication [8], chromatin modification [9], DNA damage response [10], and genomic stability [11,12,13,14]. The R-loops also play essential roles in transcriptional regulation [1,15], in which R-loops are inclined to co-localize with open chromatin and recruit transcription activation factor [16]. However, the role of R-loops in biological processes such as cell fate transition has not been fully studied.

Differentiated somatic cells can be reprogramed into induced pluripotent stem cells (iPSCs) by forced expression of Yamanaka factors Oct4, Sox2, Klf4, and c-Myc (OSKM) [17,18]. Due to the molecular features of iPSCs being similar to those of embryonic stem cells (ESCs) [19], the generation of iPSCs has a huge impact both on basic research and clinical applications [20,21]. Somatic cell reprogramming is a multilayered regulation process, in which key histone modifications (HMs) play a very important role for transcriptional regulation [22,23]. Extensive studies have shown that the activation of early transcriptional events in reprogramming are associated with extensive loss of histone H3 lysine 27 trimethylation (H3K27me3), which represents a general opening of the chromatin state [24]. Moreover, the deposition of the active histone marker H3 lysine 4 trimethylation (H3K4me3) enables a permissive chromatin environment for facilitating gene transcription [25,26]. Furthermore, a recent study revealed that R-loops act as epigenetic markers and coordinate with SOX2 in regulating reprogramming to pluripotency [27]. Moreover, R-loops have also been proven to be able to coordinate with other epigenetic marks, which not only contribute to the maintenance of pluripotency in human embryonic stem cells (hESCs), but also influence cell fate transition during multilineage differentiation [28].

Despite the fact that the important role of R-loops in gene regulation has been elucidated in detail, there is still a lack of general understanding about the formation dynamics of R-loops during reprogramming. In particular, the underlying mechanisms of R-loops driving cell fate transitions in reprogramming remain unexplored. In this study, we characterized the genome-wide formation of R-loops and found that the dynamic transition of R-loops was strongly linked to gene expression during the reprograming process. Moreover, the activation of the pluripotent transcriptional regulatory network requires the interplay between R-loops and histone modifications, in which the formation of R-loops is accompanied by the large accumulation of active histone marker H3K4me3, with a concomitant reduction in H3K27me3. In brief, our results reveal that there is collaborative action between R-loops and HMs for controlling cell fate transition during the reprogramming process.

## 2. Results

### 2.1. Stage-Specific R-loop Formation during the Reprograming Process

Based on the single-stranded DNA:RNA immunoprecipitation sequencing (ssDRIP-seq) data, we aimed to elucidate the characteristics of global R-loop formation in the process of mouse embryonic fibroblast (MEF, D0) reprogramming to iPSC, and found that R-loops showed a significant dynamic pattern during the reprogramming process. Stage-specific R-loop signals on genes were observed at each reprogramming stage (Figure 1A and Appendix A). By calculating the number of R-loop peaks and their covered genes in each indicated reprogramming point, we found that R-loop peaks were rapidly formed at the initial (D1) stage, and underwent extensive erasure in the D5 stage. Then, the R-loop peaks were gradually remodeled at the genome-wide regions until the iPSC stage reached more than 2.7 million peaks (Figure 1B). The same dynamic pattern was also observed in the number of target genes of the R-loop (Figure 1C). Next, we comparatively observed the formation preference of R-loops on different chromosomes; the results display that R-loop formation is a common event, and that their formation appears to be stabilized on different chromosomes (Figure 1D). By systematically analyzing the genome-wide R-loops in the process of reprogramming, we found that about 95% of the R-loops were mainly formed in promoter, gene body and intergenic regions. In the gene bodies in particular (including exon and intron regions), about 45% of R-loops were observed (Figure 1E), showing that R-loop formation in gene bodies is prevalent [6]. Furthermore, we observed that R-loop peaks have higher signals (FPKM, fragments per kb per million uniquely mapped reads) in the upstream regions of gene transcription start sites (TSS) and the downstream regions of transcription termination sites (TTS) (Figure 1F and Appendix A); this is consistent with the fact that R-loops play a role in all stages of gene expression, from transcription initiation to its termination [28].

Moreover, R-loops tend to occur on repetitive elements of the whole genome, and approximately 75% of R-loop peaks occupy repetitive elements during the whole reprogramming process (Figure 1G). Among them, the long-terminal repeat (LTR), long interspersed nuclear elements (LINE), short interspersed nuclear elements (SINE) and simple repeat are given priority by R-loops at each reprogramming stage. Notably, the R-loop formation on most different types of repetitive elements is stage-specific.

To comprehensively explore whether the formation of R-loop has transcription factor (TF) family specificity (Appendix A), we counted the R-loop peaks observed in different TF families during the whole reprogramming process. It was found that R-loops were more inclined to appear in the nuclear receptor (NR) family than in the C2H2 family in reprogramming. Interestingly, about 60% of E-twenty six (ETS) family members may be regulated by R-loops in the iPSC stage, but the influence of R-loops on these TF families needs further exploration.

### 2.2. The Cumulative Effect of R-loops on Gene Expression Regulation in Reprogramming

More and more evidence reveal that R-loops are widespread in prokaryotic and eukaryotic organisms [1], and play an important role in many key biological processes [1,9,15,29]. However, whether the R-loop formation could affect the gene expression dynamics during reprogramming remains unclear. To address this, we collected the transcriptome data during the reprogramming process, and hierarchical clustering analysis showed that there was a high correlation between the duplicate samples. Principal-component analysis (PCA) of gene expression profiles allowed us to outline a “reprogramming pseudo time”, which can clearly distinguish the reprogramming process into the MEF stage (D0), early reprogramming process (D1-D7) and iPSC phase (iPSC) (Figure 2A and Appendix A). By comparing the dynamic patterns of R-loop and gene expression during reprogramming, we found that the R-loop formation during reprogramming was more stage-specific than the gene expression pattern (Figure 2A), suggesting that R-loop levels and gene expression are not completely coupled in the process of reprogramming. Unlike R-loop dynamics, the genome-wide H3K4me3 and H3K27me3 signals shifted continuously during reprogramming until the specific pattern of iPSCs was formed (Appendix A). Moreover, we also noted that R-loops preferentially occurred within the upstream region of the transcription start site (TSS) compared with the other two histone modifications (H3K4me3 and H3K27me3) in this process, supporting previous observations [30] (Figure 2B).

In accordance with the above results, we next explored the relationship between R-loops and differential expression genes (DEGs) in the adjacent stages of reprogramming. The results revealed that the dynamic change pattern of R-loops within the genes was intense throughout the whole reprogramming process, whereas the prominent change wave of gene expression pattern only emerged between the iPSC and D7 (Figure 2C,D). These results suggest that the formation of R-loops may have a cumulative effect on gene expression. Among them, 58% (3689/6404) of genes including Nanog, Dppa4 and Jarid2 were up-regulated in the adjacent reprogramming stages, which was consistent with the up-regulated R-loop levels (Figure 2E and Appendix A), and 49% (3290/6668) down-regulated genes in the adjacent stages also had down-regulated R-loop signals (Figure 2F). Notably, these up-regulated genes (3689) with high R-loop levels were mainly characterized by signal pathways such as cell cycle, RNA transport and Stem cell pluripotency regulation, etc. (Appendix A), which coincided with the early event of mesenchymal epithelial transition (MET) in reprogramming. In contrast, the other down-regulated genes (3290) were mainly involved in Rap1, c-AMP and MAPK signal pathways, indicating that these genes play a part in the suppression of the lineage specific program (Appendix A).

During the whole process of MEF reprogramming to iPSC, the R-loop formation within gene regions was markedly different, and most of the formation events of R-loops had obvious stage-specific selection (Figure 2G). For example, 2% of the R-loop formation events only occurred in the MEF stage (D0) and 41% of the events had induced pluripotent stem cell (iPSC) specificity, which also explained why the kinetic pattern of R-loops was more intense in the adjacent reprogramming stages. Intriguingly, the related genes occupied by these stage-specific R-loops did not show obvious stage-specific expression in the whole reprogramming process, indicating that, in addition to R-loop formation, the expression of these stage-specific genes was also synergistically regulated by other epigenetic modifications (Figure 2G). Moreover, obvious stage-specific R-loop events were observed on these stage-specific expression genes, indicating that the expression of these genes is closely related to the stage-specific formation of R-loops (Figure 2H). Hence, the above results reveal that gene expression is related to the formation of R-loops to a certain extent, while the gene transcription process is the result of multilayered collaborative regulation.

### 2.3. The Relationship between R-loop Dynamics and Gene Expression

To investigate the relationship between R-loop dynamics and associated gene activation during the acquisition of pluripotency, we integrally analyzed the relevant transcriptome and R-loop data of cell fate transition during reprogramming. Based on the differential R-loops analysis in iPSC compared with D0 (Figure 3A), 7084 genes with differential R-loop signals in iPSC were identified, in which, 92% genes had up-regulated R-loop signals (6497/7084). Moreover, 7845 differential expression genes (DEGs) were found in iPSC, including 3767 up-regulated and 4078 down-regulated genes (Figure 3B). Correspondingly, 2886 of these DEGs were directly related to R-loops (Figure 3C), in which the strength of the gene expression (*n* = 1430) correlated positively with the R-loop levels (Appendix A). In particular, 1271 up-regulated genes with high R-loop signals were identified. Functional analysis of these genes showed that they were mainly involved in stem cell pluripotency regulation and transcription-related signaling pathways, which were essential for iPSC pluripotency regulation and establishment (Figure 3C,D). Additionally, the down-regulated genes (*n* = 159) with low R-loop signals were highly enriched in MAPK, Hippo, RAS and other signaling pathways (Figure 3C,E). These results indicate that the dynamics of R-loops are closely correlated with gene expression, and play a regulatory role in the reprogramming of MEF to iPSC.

Fibroblast reprogramming to the pluripotent state is governed by multiple epigenetic marks, whereby active epigenetic modifications direct cell identity away from the fibroblast, and repressive epigenetic marks are inherited from the fibroblasts [31]. In order to further explore the influence of different histone modifications on these genes regulated by R-loops, we also calculated the signal intensities (Figure 3F) of active histone marker (H3K4me3) and inhibitory histone marker (H3K27me3) around the TSSs (TSS ± 2 kb) of DEGs associated with the R-loops. As expected, the average H3K27me3 signals decreased within the 2 kb of TSSs of these up-regulated genes with high R-loop signals. Likewise, the TSSs of down-regulated genes associated with low R-loop levels were accompanied by lower H3K4me3 signals and higher H3K27me3 signals. In conclusion, these results suggest that promoter R-loop formation is accompanied by the enhanced recruitment or depletion of unique epigenomic signatures, which is crucial for regulating the differential expression of genes in reprogramming.

### 2.4. Sequential Activation of Pluripotency Genes Linked to R-loops during Reprogramming

To gain insight into the mechanisms underlying the dynamic activation of the DEGs observed in iPSC compared to fibroblasts (D0), we profiled the sequential expression pattern of these genes. Based on the fuzzy C-means (FCM) clustering analysis, these genes can be categorized into six distinct clusters, labeled as Clusters 1 to 6 (C1–C6) (Figure 4A). These clusters can be further assigned to five main categories of different dynamic patterns. The genes in C1 (*n* = 1817) showed a transient increase at the iPSC stage, suggesting their key roles in pluripotency regulation. Alternatively, C3 (*n* = 1619) had an instantaneous decrease at the D0 stage, indicating that these genes are related to differentiation. Furthermore, the dynamic expression pattern of C2 (*n* = 1220) had an overall decreasing trend over the reprogramming process, while C4 (*n* = 838) had a dramatic increasing trend during reprogramming. Interestingly, the genes in C5 (*n* = 1070) and C6 (*n* = 1069) had specific expression patterns with different intermediate reprogramming stages. Then, the corresponding differential R-loops of these six clusters were identified; parts of the genes in each cluster had differential R-loop signals, so these clusters were labeled as R1—R6 (Figure 4B,C). This result further implies that the sequential activation of these genes is also regulated by other epigenetic modifications besides R-loops.

As mentioned above, we further characterized the detailed dynamics of different HMs patterns (H3K4me3, H3K27me3, Bivalent and other) on these gene clusters associated with R-loops (Figure 4D and Appendix A). Different HMs were classified into “H3K4me3-only” (H3K4me3), “H3K27me3-only” (H3K27me3), “both H3K4me3/H3K27me3” (Bivalent) and “no H3K4me3 or H3K4me3” (Other) categories based on the H3K4me3 and H3K27me3 modification signals detected on the promoter region of each gene. Interestingly, these genes are more and more modified by H3K4me3 and bivalent during reprogramming (Figure 4D). In particular, the genes in R1 and R4 were preferentially modified by H3K4me3 rather than H3K27me3, while bivalent histone modifications were more likely to be co-occupied in R2 and R3 (Appendix A). The results display that there may be a varying degree of interdependency between HMs and R-loops in regulating genes transcription. 

By investigating the influence of R-loops on the dynamic expression patterns of these genes (Figure 4E and Appendix A), we found that the expression of most of these genes was positively correlated with R-loop signals, whether in transiently up-regulated gene cluster (R1) at the iPSC stage or in the gene cluster which have a continuous up-regulated trend during reprogramming (R4). For instance, the expression of most pluripotent genes (such as *Dppa2*, *Dppa3*, *Sall4*, etc.) in R1 is directly related to the regulation of R-loops, and the PCC can reach 0.7 (Figure 4E and Appendix A). Nevertheless, in the other clusters, the effect of R-loops on the expression of these genes did not show a positive or negative regulatory preference, further suggesting that the R-loops had a selective preference for gene expression regulation (Appendix A). Then we selected some representative genes from R1-R4 and found that the presence of R-loops was indeed related to the expression of genes (Figure 4F). In R1, with the instantaneous up-regulation of pluripotency genes in the iPSC stage, the R-loop signals also present a transient up-regulation pattern. All in all, these results demonstrate that there is a subset of R-loops which plays an important role in the sequential activation of pluripotency genes in reprogramming.

### 2.5. The Collaborative Regulation of R-loops and HMs Facilitates the Pluripotency Network Activation during Reprogramming

To investigate how the expression dynamics of these gene clusters are reflected in HM changes, we tracked the binding abilities of H3K4me3 (activating) and H3K27me3 (repressing) on these clusters at D0 and iPSC, respectively (Figure 5A). Interestingly, the genes in R1 and R4 enriched higher H3K4me3 and lower H3K27me3 signals in iPSC than in D0, which is in accordance with the higher expression level of these genes in iPSC, suggesting that H3K4me3 and H3K27me3 keep their antagonistic relationship in regulating gene expression during the reprogramming process. At the same time, compared with D0, we observed lower H3K4me3 levels and higher levels H3K27me3 in R2 and R3 at the iPSC stage, which coincides with the fact that these genes were inhibited in iPSC. Additionally, in the remaining two gene clusters (R5 and R6), the H3K27me3 level at D0 was higher than that in iPSC, while H3K4me3 levels showed no significant difference in D0 and iPSC. These results indicate that these gene clusters exhibit different dynamic expression patterns under the collaborative regulation of H3K4me3 (activating) and H3K27me3 (repressing).

Next, we attempted to explore the relationship between different HM patterns and R-loops on gene expression in R1 and R3 (Figure 5B). The result showed that these genes, highly expressed in iPSC (R1) or in D0 (R3), were attributed to the positive regulation of H3K4me3 and bivalent histone modifications. On the contrary, the decreased expression of these genes in D0 or iPSC was associated with the H3K27me3 epigenetic mark (Figure 5B, right). Interestingly, different HM patterns do not directly affect the R-loop modification signals on gene clusters (Figure 5B, left). To verify this hypothesis that the deposition of these histone markers may be related to the formation of R-loops, we further explored the modification differences between H3K4me3 and H3K27me3 in the formation region of R-loops in different gene clusters (R1 and R3) (Figure 5C). It was found that the gene activation in iPSC (R1) was consistent with the deposition of H3K4me3 in the R-loop formation regions, accompanied by the decrease in H3K27me3 signals. In agreement, the inhibition of genes (R3) at the iPSC stage is related to the decrease in H3K4me3 and the high H3K27me3 levels in the R-loop formation region. Finally, we mapped the dynamic transformation of the regulatory networks of pluripotent genes associated with R-loops during reprogramming, to determine whether the R-loop transition and network activation simply followed the same dynamic pattern (Figure 5D,E). The results showed that the activation of pluripotent transcriptional regulatory networks (TRNs), including *Dppa2*, *Dppa3* and *Zscan4b*, was accompanied by the dynamic transformation of R-loops. Together, the above results indicate that collaborative interactions between R-loops and HMs can facilitate the activation of the pluripotency regulation network, thereby driving cell fate transformation during reprogramming.

## 3. Discussion

The regulatory roles of R-loops in yeast, plants and animals have been extensively investigated. However, the potential effect of R-loop dynamics on cell fate transition in somatic cell reprogramming remains unclear. In this study, we profiled the genome-wide formation of R-loops during the reprograming process and found that the formation of R-loops mainly occurs in the promoter, gene body and intergenic regions. Importantly, the formation events of R-loops mainly occur in repetitive elements and the R-loop formation on these repetitive elements has stage-specificity in cell reprogramming. Additionally, R-loop occupation also has TF family specificity, in which they prefer to appear in the NR family than C_2_H_2_ family during the whole reprogramming process. However, the regulatory mechanism of R-loops on this TF family needs to be further explored.

Previous studies have indicated that R-loops as new candidate regulators play important roles in gene expression [15,27,28]. Consistently, we found that R-loop dynamics were also closely coupled with gene expression in reprogramming, and that the cumulative effect of R-loops on gene expression is extremely significant in this process. This is most easily interpreted to explain why the kinetic pattern of R-loops was intense for the entire duration of reprogramming, whereas the genes were activated on a large scale in the last stage of reprogramming. However, the gene transcription in the reprogramming process is the result of multilayered collaborative regulation [32], in which the dynamics of R-loops were accompanied by the enhanced recruitment of H3K4me3 and the depletion of H3K27me3 for regulating gene expression in this process. These results support the fact that R-loops and HMs collaboratively shape cell fate transition in reprogramming. Furthermore, the activation of pluripotent TRNs followed the same dynamic pattern with R-loop formation, further displaying that R-loops indeed play an important regulatory role for accelerating cell fate transition in reprogramming.

## 4. Materials and Methods

### 4.1. Dataset Collection

Both the single-stranded DNA:RNA immunoprecipitation sequencing (ssDRIP-seq) dataset and RNA sequencing (RNA-seq) data of mouse embryonic fibroblast (MEF) reprogramming to the iPSC state were downloaded from the Gene Expression Omnibus (GEO) database under accession number GSE125644 (https://www.ncbi.nlm.nih.gov/geo/query/acc.cgi?acc=GSE125644, accessed on 12 November 2020) [27], including six different reprogramming stages: MEF/day 0, day 1, day 3, day 5, day 7, induced pluripotent stem cells (D0, D1, D3, D5, D7, iPSC). For chromatin immunoprecipitation sequencing (ChIP-seq) a dataset of six different reprogramming stages was also downloaded from GEO database and GEO accession no. GSE67520 (https://www.ncbi.nlm.nih.gov/geo/query/acc.cgi?acc=GSE67520, accessed on 13 November 2020) [20], which included two histone modifications of H3K4me3 and H3K27me3.

### 4.2. ssDRIP-seq Data Processing

The ssDRIP-seq raw reads were filtered by Cutadapt software (version 2.10, EMBnet.journal, Uppsala, Sweden) [27,33] and reads less than 50 bases in length were discarded. Trimmed reads were mapped to the mouse genome assembly mm10 by using Bowtie2 (version 2.2.9, Nature Methods, New York, NY, USA) short read alignment software with default parameters. Then, R-loop peak calling was performed using MACS2 (version 2.1.0) [34,35,36] using the following parameters: macs2 callpeak −t $treatmentsam −c $controlsam −f SAM −keep−dup 1 −*n* $name −g 1.87e9 −B −q 0.01. Finally, R package ChIPseeker [37] was used to annotate with the position of the peaks in the genome, in which −2 kb to 1 kb of gene transcription start site (TSS) were defined as gene promoter. We also calculated the occupancy of each R-loop as FPKM (fragments per kb per million uniquely mapped reads) for their signals. Genome-wide R-loop signals were calculated from absolute mapped read density over 300 bp sliding windows using ‘ngs.plot.r’ from ngsplot (version 2.63, BMC Genomics, Basingstoke, UK) [38].

### 4.3. ChIP-seq Data Processing

For ChIP-seq data processing, the original data were controlled by Cutadapt software to remove low-quality reads. Next, the clean reads were aligned to mm10 by using Bowtie2 (version 2.2.9) with default parameters [39]. Then we used MACS2 (version 2.1.0, Current Protocols in Bioinformatics, Hoboken, NJ, USA) (with the parameters setting: macs2 callpeak −t $treatmentsam −c $controlsam −f SAM −keep-dup 1 −n $name −g 1.87e9 −B −q 0.01) to call binding peaks [34,35]. The ChIP-seq coverage densities of H3K4me3 and H3K27me3 were plotted by mapping reads to regions that were 2-kb up/downstream of the TSS for each gene, respectively. Each gene region was split into 300 equally sized bins. The average ChIP-Seq density of H3K4me3 and H3K27me3 in the R-loop regions were calculated using ngsplot (version 2.63) [38].

### 4.4. RNA-seq Analysis

The pair-end raw RNA-seq reads were trimmed by the Trimmomatic (version 0.38, Bioinformatics, Oxford, UK) [40] and mapped to mm10 reference genome using Hisat2 (version 2.1.0, Nature Protocols, New York, NY, USA) aligner with default parameters [41]. The retained reads were subsequently assembled by using Stringtie (version 1.3.5, Nature Biotechnology, New York, NY, USA) [41,42]. To eliminate the effects of sequencing depth and transcript length, normalized FPKM (fragments per kilobase of exon model per million mapped reads) for each gene were conducted. Read counts for each gene were calculated using HTseq (version 0.11.0, Bioinformatics, Oxford, UK) [43].

### 4.5. Identification of R-loop Peaks on Repetitive Elements and Transcription Factor (TF) Family

For alignments to repetitive regions in the genome, repeat annotations were downloaded from the University of California at Santa Cruz (UCSC) browser (RepeatMasker, mm10) [44]. R-loop peaks were assigned to TF families according to the JASPAR database [45]. TF families with fewer than two members were grouped under “Other.”

### 4.6. Differential Gene Expression Analysis

Differential expression analysis was performed by R package DEseq2 [46,47]. For each comparison, genes with a Benjamini and Hochberg-adjusted *p* value (false discovery rate, FDR) < 0.05 and the absolute of Log2(fold change, FC) > 1 were regarded as differential expression genes (DEGs) [48,49,50].

### 4.7. Fuzzy C-Means (FCM) Clustering Analysis

Fuzzy cluster analysis was performed by using mFuzz [51] for cluster analysis of the gene expression pattern, the normalized expression levels of all DEGs between iPSC and D0 were analyzed.

### 4.8. Transcriptional Regulatory Networks Analysis

Transcriptional regulatory network (TRNs) analysis of representative genes associated with R-loops (as shown in Figure 5D) were implemented by STRING database (version 11.0) [52]. Then, the TRNs were visualized by Cytoscape (version 3.7.0, Genome Research, New York, NY, USA) [53,54]. In TRNs, the color indicates R-loop signals and box size indicates gene expression levels.

### 4.9. Functional Enrichment and Statistical Analysis

Kyoto Encyclopedia of Genes and Genomes (KEGG) pathway enrichment analysis was performed based on the R package clusterProfiler (version 3.14.3, OMICS: A Journal of Integrative Biology, New York, NY, USA) [55]. Statistical analyses were implemented with R (version 3.6.0, http://www.r-project.org, accessed on 1 June 2020). Representative KEGG pathways with *p* values < 0.05 were summarized in each gene cluster. The Pearson correlation coefficients (PCC) between biological replicates were calculated using the ‘cor’ function with default parameters [56,57]. A Student’s *t* test was performed using the ‘t.test’ function with default parameters, and *p* values < 0.05 were considered statistically significant [58,59].

## 5. Conclusions

Taken together, our results not only profiled the unique dynamic patterns of R-loops in reprogramming, but also deciphered that a crosstalk exists between R-loops and HMs to regulate cell fate transition during the reprogramming process.

## Figures and Tables

**Figure 1 ijms-23-01567-f001:**
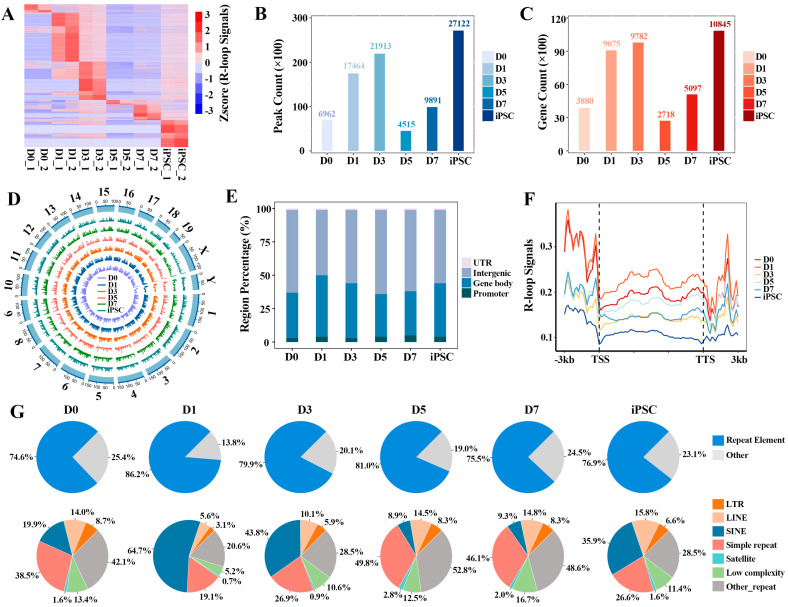
Genome-wide identification of R-loops during reprogramming: (**A**) heatmap displaying the global R-loop signals (FPKM) on genes at different reprogramming stages; (**B**) the number of R-loop peaks at different reprogramming stages; (**C**) the target genes’ number of R-loop peaks at different reprogramming stages; (**D**) Circos plot showing the R-loop signals across all the chromosomes in reprogramming; (**E**) the distribution of R-loop peaks on different genomic features; (**F**) the average distribution of R-loop signals at 3 kb region flanking of TSS and transcription termination sites (TTS) during reprogramming progress; (**G**) pie plot showing the distribution of R-loop peaks in various types of repetitive elements.

**Figure 2 ijms-23-01567-f002:**
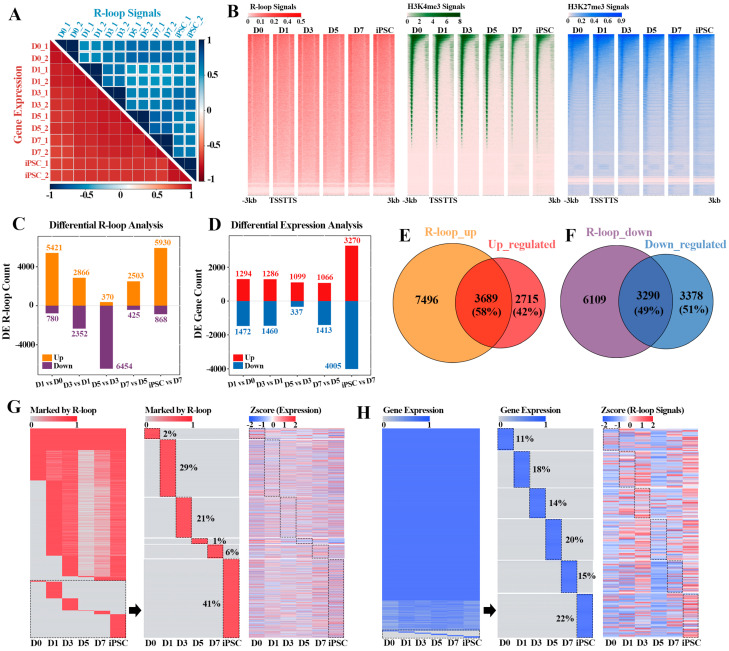
Reprogramming stage-specific R-loop modification and gene expression: (**A**) correlation heatmaps of different reprogramming stages according to R-loop signals (blue—top heatmap) and gene expression levels (red—bottom heatmap). Color scales represent Pearson correlation coefficients (PCC) between different stages; (**B**) heatmaps displaying the signals of R-loop, H3K4me3 and H3K27me3 in flanking TSSs at different reprogramming stages. In each panel, each row represents a 3 kb region flanking TSS; (**C**) analysis of differential R-loop signals in adjacent stages of reprogramming; (**D**) analysis of differential gene expression in adjacent stages of reprogramming; (**E**) Venn diagram revealing the overlap between genes with up-regulated R-loop signals (as shown in upper bar of (**C**)) and up-regulated genes (as shown in upper bar of (**D**)) between adjacent reprogramming stages; (**F**) Venn diagram showing the overlap between genes with down-regulated R-loop signals (as shown in lower bar of (**C**)) and down-regulated genes (as shown in lower bar of (**D**)) between adjacent reprogramming stages; (**G**) heatmaps showing the genes marked by R-loop peaks (left) during reprogramming progress, part of genes marked by R-loop peaks with reprogramming stage-specificity (middle) and the expression levels corresponding to these genes (right), respectively. Red—R-loop peaks; gray—no R-loop peaks; (**H**) heatmaps showing the dynamic expression profiles (left) of genes during reprogramming progress, part of gene expression with reprogramming stage-specificity (middle) and the R-loop signals corresponding to these genes (right), respectively. Blue—expression levels > 0; gray—no expression.

**Figure 3 ijms-23-01567-f003:**
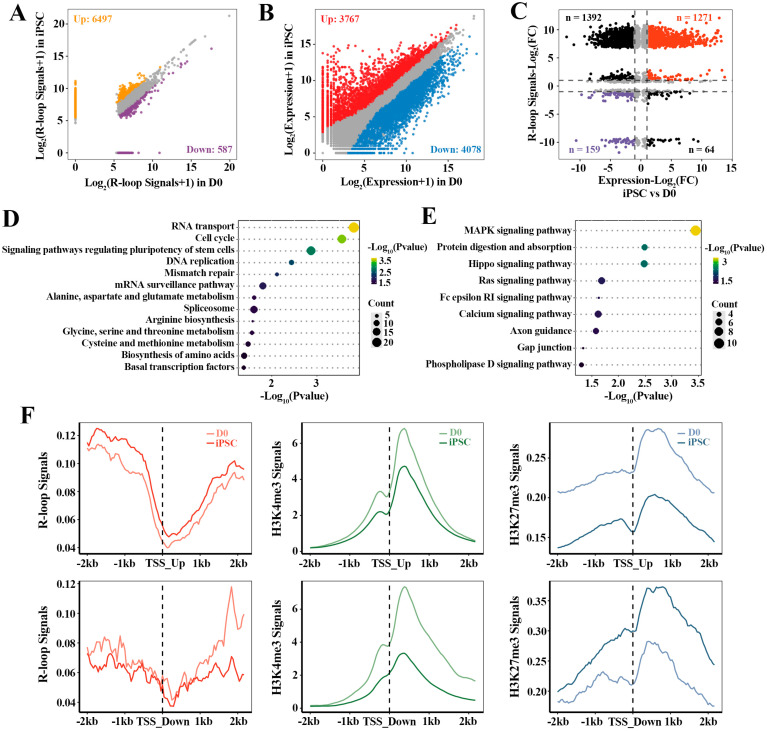
The dynamic effect of R-loops on gene expression in reprogramming: (**A**) analysis of differential R-loop signals in iPSC vs. D0; (**B**) analysis of differential gene expression in iPSC vs. D0; (**C**) scatterplots showing the relationship between R-loops and gene expression during reprogramming process. Red dots are the genes whose expression levels and R-loop signals are up-regulated in iPSC compared to D0. Purple dots are the genes whose expression levels and R-loop signals are down-regulated in iPSC; (**D**,**E**) KEGG pathway analysis of up-regulated genes with higher R-loop signals (*n* = 1271) and down-regulated genes with lower R-loop signals (*n* = 159); (**F**) the read density profiles of R-loops, H3K4me3 and H3K27me3 at the 2 kb flanking TSSs of the up-regulated genes (*n* = 1271, upper three panels) and down-regulated genes (*n* = 159, lower three panels).

**Figure 4 ijms-23-01567-f004:**
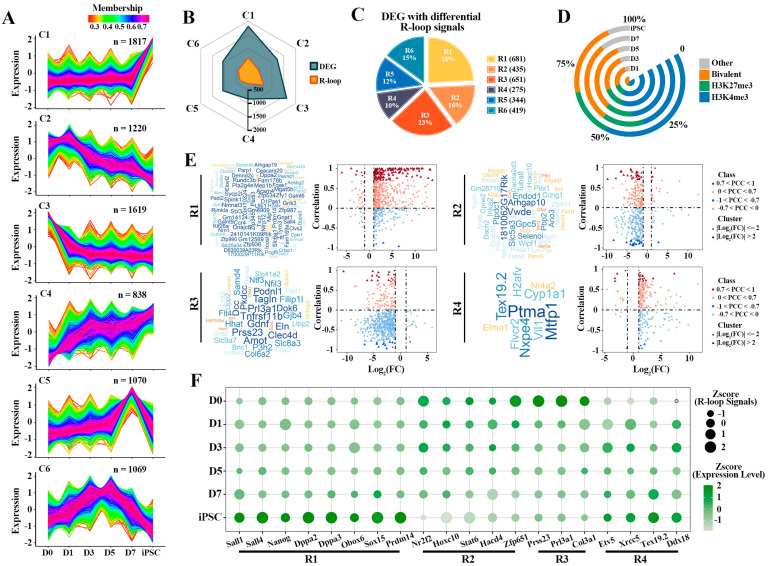
Correlation of R-loop levels with gene expression of different clusters: (**A**) fuzzy C-means (FCM) clustering analysis according to the expression profiles of DEGs. Average scaled FPKM values are represented by each line. Colors indicate the membership value of each gene in the current cluster; (**B**) the radar chart displaying the overlap of genes with differential R-loop signals and six different clusters; (**C**) the percentage of DEGs with differential R-loop signals in each cluster is indicated. These clusters have differential R-loop signals, labeled as R1-R6; (**D**) proportion of co-differential genes (the genes involved in R1-R6) bound by “H3K4me3-only” (H3K4me3—blue), “H3K27me3-only” (H3K27me3—green), “both H3K4me3/H3K27me3” (Bivalent—orange) and “no H3K4me3 or H3K4me3” (Other—grey) modification; (**E**) integrated analysis of DEGs and differential R-loop signals in R1—R4. Word clouds represent the key genes (0.7 < |PCC| < 1; |Log2(FC)| > 2) in each group, both size and color of the word are proportional to the membership value of each gene. Y-axis of the scatter plot indicates the correlation between expression levels and R-loop signals, X-axis indicates logarithmic transformation of the fold change (FC) of the expression level (iPSC vs. D0); (**F**) dynamic changes of gene expression levels and R-loop signals of representative genes in each cluster. The dot sizes indicate expression levels and colors display R-loop signals.

**Figure 5 ijms-23-01567-f005:**
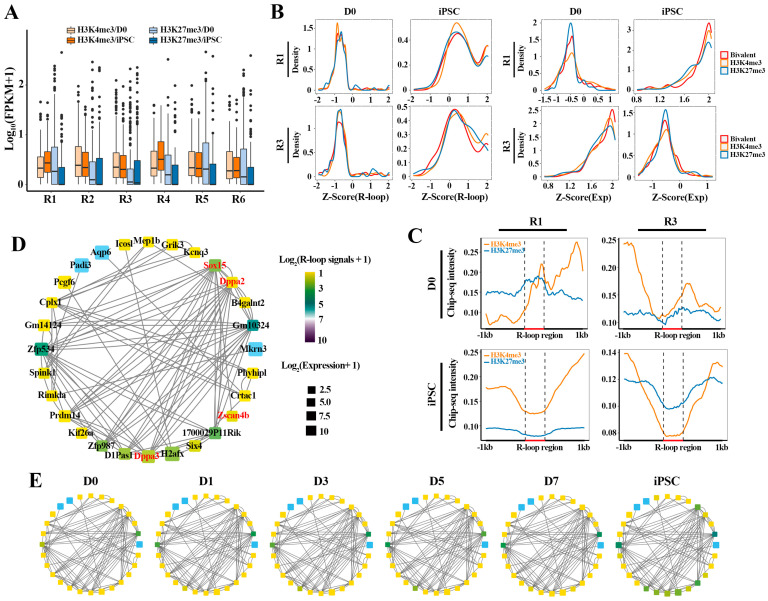
Dynamic transition of TRNs related to R-loops and HMs in reprogramming: (**A**) boxplot showing H3K4me3 and H3K27me3 binding signals (FPKM) on genes associated with bivalent histone modifications in D0 and iPSC; (**B**) the density distribution of R-loop signals (FPKM) and expression levels for genes (involved in R1 and R3, respectively) bound by H3K4me3, H3K27me3 and bivalent histone modifications; (**C**) average H3K4me3, H3K27me3 signals in 1 kb flanking R-loop regions for R1 (*n* = 681) and R3 genes (*n* = 651) at D0 and iPSC stages. These R-loops were located in promoter regions of genes; (**D**,**E**) dynamic transition of transcriptional networks from MEF (D0) to iPSC in R1. Color indicates R-loop signals and box size indicates gene expression levels. Genes symbols as shown in (**D**).

## Data Availability

The ssDRIP-seq and RNA-seq datasets of different reprogramming points were downloaded from Gene Expression Omnibus (GEO) database under accession number GSE125644 [27]. The ChIP-seq data of histone modification (GSE67520) was reanalyzed in this study [20].

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
