# Peer review of "The Cumulative Formation of R-loop Interacts with Histone Modifications to Shape Cell Reprogramming"

_ijms, 2022, doi:10.3390/ijms23031567_

Round 1

Reviewer 1 Report

Thank you for the opportunity to review the manuscript: “The Cumulative Formation of R-loops Interplay with Histone 2 Modifications to Shape Cell Reprogramming". The topics is very interesting, however the presentation of bioinformatics results have to be improved remarkably before publication and all data must be available at least in the supplementary materials.

Major points:

  1. The quality of the Figures is poor. Figure 1 looks like poster – panels A-G, but the size of these mini sub-panels are so small that is not possible to check results. Very often is the result described in ONE sentence!?! It must be elaborated and described in detail with the link to the complete datasets of performed analyses and results.
  2. The same for Figures 2 – 8 panels A-H – not readable in A4 format and Figures 3-4. Probably good for poster but not for the presentation in the paper – the links to the source data are missing – so it is impossible to check and verifiy results. I used 800% magnification for Fig. 4E – still it is not readable … letter blurred – It must be readable on the A4 page.
  3. The same is true for Supplementary. Moreover – results of their analyses are not available in any readable and editable format.

Due to pure presentation and missing data - I was not able to evaluate the manuscript quality. Results of analyses must be provided.

Author Response

Point 1: The quality of the Figures is poor. Figure 1 looks like poster – panels A-G, but the size of these mini sub-panels are so small that is not possible to check results. Very often is the result described in ONE sentence!?! It must be elaborated and described in detail with the link to the complete datasets of performed analyses and results. 

Response 1: Thank you for your professional comments. Based on your suggestions, we have improved the quality of Figure 1 and added more details to describe the results (See Lines 77-84). At the same time, complete datasets related to the analysis results were also added to the Supplementary materials in the revised manuscript (See Supplementary Table S1 and S2). Moreover, we reordered the genes in Fig. 1A according to the stage specificity of R-loop signals to better display the analysis results (See revised Fig. 1A).

Point 2: The same for Figures 2 – 8 panels A-H – not readable in A4 format and Figures 3-4. Probably good for poster but not for the presentation in the paper – the links to the source data are missing – so it is impossible to check and verifiy results. I used 800% magnification for Fig. 4E – still it is not readable … letter blurred – It must be readable on the A4 page.
Response 2: Thank you for your professional comments. Based on your suggestions, we have improved the quality of Figures 2-4 to increase the readability of these figures on A4 page. In this study, the source datasets were from Gene Expression Omnibus (GEO) public database and the links to the source data also been added in the revised manuscript (See Materials and Methods-Lines 377 and 382; Data Availability Statement). To better display the results of Fig. 4E, we added complete processed data into revised manuscript (See Supplementary Table S4).

Point 3: The same is true for Supplementary. Moreover – results of their analyses are not available in any readable and editable format.

Response 3: Thank you for your professional comments. Based on your suggestions, the quality of Supplementary Figures has been improved to increase the readability and the complete processed data has been added into revised manuscript (See Supplementary Table S4).

We feel sorry for the inconvenience brought to you. Based on your suggestions, the size of sub-panels in Figures 1-4 and Supplementary Figures have been enlarged to increase the readability. And the editable tables related to results have been added into Supplementary materials for better support the results. We sincerely hope that the revised manuscript will enable you to better check the results.

Reviewer 2 Report

I have reviewed the manuscript, in which author studied the genome-wide formation of R-loops during cell reprogramming. Author examined that the dynamic transition of R-loops strongly linked to the gene expression during the reprograming process. Furthermore, author mentioned that the activation of pluripotent transcriptional regulatory network requires the interplay between R-loops and histone modifications, in which the formation of R-loops accompanied by the large accumulation of active histone marker H3K4me3, with a concomitant reduction of H3K27me3.

The work seems to fall in line with the journal’s scope, providing insights on collaborative action between R-loops and HMs during the reprogramming process. Therefore, I recommend that this manuscript is ready for the publication as it is.

Round 2

Reviewer 1 Report

Thank you. Authors have improved the manuscript and added results of their analyses in supplementary materials. However, some interpretation issues and minor changes should be updated before publication, therefore I recommend minor revision.

  1. Lines 91-97, you claim that 45% of R-loops are in „gene bodies“. But in the next setences is that majority of the R-loops are before TSS site (?) – line 102 – „In this analysis, we found that approximately 75% of R-loop peaks occupied in repetitive elements“ . What you mean by „gene bodies“ – you mean genes including or excluding promotor regions? The % do not correspond (?). If the majority of the R-loop are in „genes“ – why is more R-loops in promoters and 75% of peaks in repetitive elements? Please explain clearly in the text.
  2. There is any reason to show nine decimal places at supplementary files? Reduce the number of displayed decimal places for clarity.

Author Response

Response to Reviewer 1 Comments

Point 1: Lines 91-97, you claim that 45% of R-loops are in „gene bodies“. But in the next setences is that majority of the R-loops are before TSS site (?) – line 102 – „In this analysis, we found that approximately 75% of R-loop peaks occupied in repetitive elements“ . What you mean by „gene bodies“ – you mean genes including or excluding promotor regions? The % do not correspond (?). If the majority of the R-loop are in „genes“ – why is more R-loops in promoters and 75% of peaks in repetitive elements? Please explain clearly in the text.

Response 1: Thank you for your professional comments. In our studies, gene body including exon and intron regions and the definition of the gene body regions has been added in the revised manuscript (See Line 90). Based on your suggestions, the results have been explained clearly in the manuscript (See Lines 91-103). The detail description are as follows: By systematically analyzing the genome-wide R-loops in the process of reprogramming, we found that about 95% of the R-loops were mainly formed in promoter, gene body (including exon and intron regions) and intergenic regions. Especially in the gene body regions, about 45% of R-loops were observed (Figure 1E), showing that the R-loop formation in gene bodies is prevalent [6]. And we observed that R-loop peaks have higher signals (FPKM, fragments per kb per million uniquely mapped reads) in the upstream regions of gene transcription start site (TSS) and the downstream regions of transcription termination sites (TTS) (Figure 1F and Supplementary Table S2), consistent with the fact that R-loops play a role in all stages of gene expression from transcription initiation to its termination. Moreover, R-loops tend to occur on repetitive elements of the whole genome and approximately 75% of R-loop peaks occupy in repetitive elements during the whole reprogramming process (Figure 1G).

Point 2: There is any reason to show nine decimal places at supplementary files? Reduce the number of displayed decimal places for clarity.

Response 2: Thank you for your professional comments. Based on your suggestions, the number of displayed decimal places (three decimal places are reserved) in Supplementary files has been reduced to display the result clearly (See Supplementary Tables).